# Geospatial mapping to assess the distribution and determinants of zero dose vaccination status hot spots among children in Ethiopia using EDHS 2019: Spatial and geographical weighted regression

**Muluken Chanie Agimas**[1]*, **Meron Asmamaw**[1], **Melese Kebede Hailu**[2], **Tigabu Kidie**[1], **Habtamu Wagnew Abuhay**[1], **Getaneh Awoke Yismaw**[1], **Nebiyu Mekonnen Derseh**[1]

1 Department of Epidemiology and Biostatistics, Institute of Public Health, College of Medicine and Health Science, University of Gondar, Gondar, Ethiopia, 2 Department of Emergency and Critical Care Nursing, College of Health Sciences, Debre Tabor University, Debre Tabor, Ethiopia

* mulukensrc12@gmail.com

## Abstract

### Introduction

Zero-dose vaccine status refers to a child who has not yet received any childhood vaccines. Globally, zero-dose vaccination status is the major public health problem. In sub-Saharan African countries, among five children, one child did not access the vaccines. The efforts to identify the factors contributing to the zero-dose vaccine have not yet been addressed in Ethiopia.

### Objectives

To assess the distribution and determinants of zero dose vaccination status hot spots among children in Ethiopia using Ethiopian demographic health survey 2019.

### Methods

A secondary analysis of a cross-sectional study was used among a total of 3208 participants. Stata-14, Arc geographic information system-10.3, and Sat Scan software were used to analyze the data. Moran's I statistic was used to assess the non-random distribution of zero-dose vaccination status. Spatial and geographically weighted regression was used to map the distribution and determinants of zero-dose vaccination status in Ethiopia. The Getis-Ord-statistics and Sat Scan analysis were also used to identify the hot spot area and significant clusters respectively. To identify significant factors, a p-value <0.05 was used.

### Results

The prevalence of zero-dose vaccination status among children aged 12–35 months old was 16.3% (95%CI, 15%–17.6%). The distribution of zero-dose vaccination status in

**Data Availability Statement:** All relevant data are within the manuscript and its Supporting Information files.

**Funding:** The author(s) received no specific funding for this work.

**Competing interests:** The authors have declared that no competing interests exist.

**Abbreviations:** AIC, Akaike information criteria; ANC, Antenatal Care; DHS, Demographic Health Survey; EAs-. . . . ., Enumeration Areas; EDHS, Ethiopian Demographic Health Survey; GWR, Geographic Weighed Regression; LLR, Log Likelihood Ratio; OLS, Ordinary Least Square; RR, Relative Risk; SNNPR, South Nation and Nationality of People Representative; VPDs, Vaccine Preventable Diseases.

Ethiopia was not randomly distributed across the area (Moran's I value = 0.4, p<0.001). The hotspot area of zero-dose vaccination status in Ethiopia was located in Afar, Somalia, and the south nations region. The primary cluster was located at 5.479641 N and 42.196835 E within a 405.96 km radius in the Somali region. The highest effect (β, 0.23–0.24) of give birth at home on zero-dose vaccination status was observed in Afar, Somalia, and Tigray, whereas the lowest effect of home delivery (β, 0.21–0.22) was observed in south nations, Gambela, and the western part of Benishangul Gumuz. Additionally, the poor wealth index had the highest effect on zero-dose vaccination status (β, 0.06–0.07) in south nations, Benishangul Gumuz, and the Gambela region, and the lowest effect of the poor wealth index (β, 0.03–0.04) was observed in Somalia and the Afar region.

## Conclusion

The zero dose child in Ethiopia was low. Geographically, the highest proportion of zero-dose vaccination status among children was in the Somali and Afar regions. Home delivery had the highest and most positive effect on zero-dose vaccination status in Afar, Somalia, and the Tigray region. Additionally, the poor wealth index had the highest and most positive effect in Somalia and the Afar region. Expanding delivery at the health facility and economic empowerment of women are recommended to improve vaccination in Ethiopia.

## Introduction

Zero-dose vaccine status refers to a child who has not yet received any childhood vaccines [1]. Globally, each year, about 2.5 million cases of child mortality can be prevented using childhood vaccination [2, 3]. In sub-Saharan African countries, among five children, one does not access the vaccines [4]. Consequently, in each year, 30 million and half million less than five-year-old African children are affected by vaccine-preventable diseases and die, respectively [5]. According to the estimations of the World Health Organization and the United Nations Children's Fund, in low-and middle-income countries, about 14 million children in 2019 and 17 million under five children in 2020 did not receive any vaccine doses [6]. This indicated that in low and middle-income countries, the numbers of under-5 children with zero-dose children increased across time.

An increment of a zero-dose vaccine could be an indicator of not achieving the national vaccine coverage goal over time in sub-Saharan Africa (SSA) (90%) [7]. To overcome this problem, the World Health Organization (WHO) set a strategy goal to reduce the number of zero-dose children by 50% in 2030 [8]. Additionally, by 2030, the sustainable development goals target to zero the mortality of children [9]. To achieve this goal, evidence-based strategy and policy are very important to reduce zero-dose children and vaccine-preventable diseases (VPDs) [10]. Childhood disability is also reduced by the zero tolerance of zero-dose children [11–13]. Particularly, lack of attention for zero-dose children, rural residence, and low educational status are the contributing factors for zero-dose children [14–16]. Because of zero-dose children, the reemerging of eradicated diseases has been a threat to public health [17–20]. Childhood vaccination is the most cost-effective strategy for VPDs like poliomyelitis, measles, pneumonia, hepatitis B virus, diphtheria, Haemophilus influenza type B (Hib), tuberculosis, diarrhea, and others [21, 22]. According to the EDHS, between 2010 and 2020, in sub Saharan African countries the coverage of zero dose children was 9.1% [23]. In Ethiopia between, 2000

and 2019, the basic vaccination coverage had progressed from 14.3% to 44.1%. The vaccination coverage was estimated to reach 53.6% by 2025, indicating a significant gap in achieving the intended target of 90% [24]. There is a global strategic goal from 2021–2025 to reach zero-dose children and miss opportunities for routine immunization [25–27]. But the Corona virus pandemic was a threat to the immunization program, which increased the number of zero-dose children by 37% [28]. The efforts of identifying the factors of the zero-dose vaccine are not yet addressed, and therefore studies are needed to assess the prevalence and determinants of zero-dose children in Ethiopia. Additionally, to apply specific and cost-effective interventions, the spatial and geographical weighted regression is very important to show the geographical distribution of zero-dose children in Ethiopia and the effect of each predictor across the area. But nothing is known in Ethiopia. Therefore, this study aimed to assess the geographical distribution and apply geographical weighted regression to identify the determinants of zero-dose vaccination status hot spots among children in Ethiopia using EDHS 2019.

## Methods

### Study design, area and period

The study used the secondary analysis of the EDHS-2019 community-based cross-sectional study conducted from March 21 to June 28, 2019. Ethiopia is a low-income country located in East Africa. The lowest latitude of Ethiopia is Dallol (128 meters above sea level), which is found in Afar Ethiopia, and the maximum latitude is located in Ras Dashen (4620 meters above sea level). Amhara Ethiopia [29]. Currently, it comprises 12 regional states, namely Amhara, Tigray, Afar, Benishangul Gumuz, Afar, Somalia, Sidama, Oromia, Gambela, Harari, South Ethiopia, Southwest Ethiopia, and central Ethiopia. Additionally, Ethiopia has two self-governed city administrations: Addis Ababa and Diredawa. As of Wednesday, December 27, 2023, according to the Worldometre estimate, the total population of Ethiopia is 128,073,400, and 22.1% of the population is urban [30].

### Population

All women who had children in Ethiopia were the source population, and all women who had children aged 12–35 months in the enumeration area were the source population and included in the study.

### Data source

The data source for the current study was the 2019 EDHS, which is the fifth DHS implemented in Ethiopia. It is conducted by the Ethiopian public health institute with the collaboration of the Ethiopian statistical agency and the Federal Ministry of Health. The EDHS aimed to measure the progress of the health sector goals set under the Growth and Transformation Plan (GTP), which is closely aligned to the Sustainable Development Goals (SDG). The survey was conducted among reproductive-age women on the following characteristics: background characteristics, fertility determinants, marriage, awareness and use of family planning methods, child feeding practices, nutritional status of children, childhood mortality, and pregnancy care received from skilled providers, such as doctors, nurses/midwives, health officers, and health extension workers (ANC). The ANC care was delivered for women aged 15–49 who had a live birth in the 5 years before the survey. The survey report represents the comprehensive outcome of the nine regions of Ethiopia and two city administrations.

## Variables

**Dependent variable.**   Zero dose vaccine status (Yes, No).

**Women related independent variables.**   Wealth index, educational status, place of delivery, religion, age of the women, ANC visit, media exposure, region, current breast feeding, and current pregnancy.

## Operational definition

**Zero dose vaccine status.**   In this study, children who have not yet received any childhood vaccines are categorized as zero-dose children (yes), and children who have received at least one dose of vaccine are classified as non-zero-dose vaccines (no) [1].

**Media exposure.**   Was assessed based on whether people had access to read newsletters, listen to the radio, and watch TV. Accordingly, if they have access to all three media (newsletter, radio, and TV) at least once a week, we categorized them as "yes", otherwise "no" [31].

## Sampling method and procedure

The sampling frame for EDHS 2019 was all census enumeration areas (149,093 EAs), obtained from the central statistical agency. An average of 131 households were incorporated into each EA. The EDHS 2019 sample was stratified and selected in two stages. Each region was stratified by urban and rural areas, with a total of 21 sampling strata. A total of 305 EAs, 93 EAs in urban areas, and 212 EAs in rural areas, were selected using proportional EA size allocation techniques. In the selected EAs, household listings were conducted. Then 30 households were selected per cluster using equal-probability systematic selection techniques. A two-stage cluster sampling method was used to select 3028 women (participants) who had children aged 12–35 months in the EAs (**S1 Data**). The detailed section is reported in EDHS 2019 [32].

## Data collection and quality assurance

We used the secondary analysis of the EDHS 2019 data set. This data was collected using a pre-tested structured interview technique from March 21 to June 28, 2019. The location of the data was also collected using a geographic positioning system (2 kilometers for urban clusters and 5 kilometers for rural clusters). To assure the quality of the data, pretesting and training for data collectors and supervisors were conducted. The detail section has been reported in EDHS 2019 [32]. For the purpose of further analysis, data was requested online from the demographic health survey international at DHS's official website, www.measuredhs.com. Then the data was accessed after 2 working days. After the data was accessed, data cleaning, recoding, and overall data management were conducted.

## Data management and analysis

**Spatial analysis.**   Before further analysis, data cleaning and recoding were conducted using STATA software version 14. Sampling weight was also used to analyze the weighted frequency of all the data. After weighting the dependent and independent variables, the data was exported to Excel with a cluster and coordinate file. Then the Excel file was saved in the form of a "csv" file and exported to ArcGIS version 10.3. Using this software, the autocorrelation was assessed by Moran's I statistics with a p-value <0.05. The Moran's I statistic value ranges from 1 to -1. When Moran's I value is close to -1, it implies that the distribution of zero-dose vaccination status was dispersed across the clusters, whereas its values of 1 and 0 showed that the distribution of zero-dose vaccination status was clustered and random, respectively [33]. The hot/cold spot area of zero-dose vaccination status in Ethiopia was analyzed by the Getis-

Ord statistic, and to predict the zero-dose vaccination status in the unsampled area, we used the ordinary Kriging spatial interpolation technique. The statistically significant clusters of zero-dose vaccination status in Ethiopia were analyzed using Kuldorff Sat Scan version 9.6 software with a maximum cluster size per population of 50% [34].

## Spatial regressions

**Ordinary least squares regression.** After we mapped the distribution of zero-dose vaccination status in Ethiopia, spatial regression analysis was used to assess the effect of the predictors on zero-dose vaccination status across the clusters and areas of Ethiopia. The assumptions of ordinary least squares regression (OLS), such as no multicollinearity, non-stationary, the residual having no spatial pattern, a model having key predictors, and the residual being free from spatial autocorrelation [35–37] were checked, and all assumptions were fulfilled. The beta coefficient should be significantly associated with zero-dose vaccination status. The explanatory regression and data mining tools were used to identify the best $R^2$ value, and a model fulfilled all assumptions [35]. The final model was validated by internal cross-validation. The mean error of the model must be close to zero, and the root-mean-square standardized error must be close to one (20). Thus, all the assumptions mentioned above were satisfied.

**Geographically weighted regression.** In the OLS analysis, the effect of predictors was non-stationary across the clusters. In this case, GWR is better than OLS. The OLS is the global estimate, whereas the GWR is a local estimate. The GWR allows visualization of stimulus-response relationships and how that relationship varies in space. It also accounts for the spatial autocorrelation of variables. Using GWR, we can fit one linear regression equation, but not by OLS. The OLS cannot consider the different coefficients for each cluster. But using GWR, we can consider different coefficients for each cluster. The GWR model [38] can be written as:

$$yi = \beta0(\text{ui vi}) + \sum_{k=1}^{p}(\beta\text{k(ui vi)xik} + \varepsilon\text{i})$$

Where:
yi = observation of response
(uivi) = latitude and longitude
βk (uivi) (k = 0, 1,. . . p,) are p unknown functions of geographic locations (uivi).
xik = independent variables at location (uivi), i = 1,2,. . .n
εi = residuals with zero mean and homogeneous variance σ2.

## Ethical declaration

**Ethical approval.** Ethical approval from the participant was not applicable to the study participants rather informed consent and approval of the data access and use were obtained from the IRB board members of the Demographic and Health Surveys (DHS) Program. We downloaded the datasets for this current study from the website at http://www.dhsprogram. com. Datasets obtained from the Measure DHS program are kept confidential.

## Results

## Characteristics of the participants

Of the total 3028 women who had children aged 12–35 months old, the majority, 2442 (76.1%), were rural residents. About 2316 (72.2%) and 1447 (45.1%) of the women had a

**Table 1. Characteristics of the participants among women who had child aged 12–35 months old in Ethiopia using EDHS 2019.**

| Variable | Category | Weighted frequency | % |
|---|---|---|---|
| Wealth index | Poor | 1594 | 49.70 |
| | Middle | 449 | 14 |
| | Rich | 1,165 | 36.30 |
| Residence | Urban | 766 | 23.90 |
| | Rural | 2442 | 76.10 |
| Educational status | No education | 1648 | 51.40 |
| | Primary | 1080 | 33.70 |
| | Secondary | 296 | 9.20 |
| | Higher | 184 | 5.70 |
| Religion | Orthodox | 929 | 29 |
| | Catholic | 18 | 0.60 |
| | Protestant | 588 | 18.30 |
| | Muslim | 1633 | 50.90 |
| | Traditional | 32 | 1 |
| | Other | 8 | 0.20 |
| Region | Tigray | 261 | 8.10 |
| | Afar | 371 | 11.60 |
| | Amhara | 294 | 9.20 |
| | Oromia | 398 | 12.40 |
| | Somali | 321 | 10 |
| | Benishangul Gumuz | 289 | 9 |
| | SNNPR | 360 | 11.20 |
| | Gambela | 247 | 7.70 |
| | Harari | 251 | 7.80 |
| | Addis Ababa | 180 | 5.60 |
| | Diredawa | 236 | 7.40 |
| Age | 15–24 years | 931 | 29 |
| | 25–34 years | 1,719 | 53.60 |
| | ≥ 35 years | 558 | 17.40 |
| ANC visit | Yes | 892 | 27.80 |
| | No | 2316 | 72.20 |
| Place of delivery | Home | 1447 | 45.10 |
| | Health facility | 1761 | 54.90 |
| Media exposure | Yes | 1561 | 48.60 |
| | No | 1647 | 51.40 |
| Delivery by cesarean section | No | 3001 | 93.50 |
| | Yes | 207 | 6.50 |
| Currently breast feed | Yes | 2376 | 74.10 |
| | No | 832 | 25.90 |
| Current pregnant | Yes | 316 | 9.90 |
| | No | 2892 | 90.10 |

history of ANC visits and giving birth at home, respectively. Furthermore, 1719 (53.6%) of the women were in the age range of 25 to 34 years, and 1594 (49.7%) of them had a poor wealth index (**Table 1**).

### Prevalence of zero dose vaccination status in Ethiopia

The prevalence of zero-dose vaccination status among children aged 12–35 months old was 16.3% (95%CI, 15%–17.6%).

### Spatial analysis

**Spatial autocorrelation and distribution of zero dose vaccination status.** To map the spatial distribution and autocorrelation of zero dose vaccination status among children in Ethiopia, a total of 305 clutters were used. Thus, as Moran's I statistics showed, the distribution of zero dose vaccination status among children in Ethiopia was not randomly distributed across the area (Moran's I value = 0.4, p<0.001, z-score = 8.89) (**Fig 1**). This implies that the assumption of further spatial analysis has been fulfilled. So the highest distribution of zero-dose vaccination status was in the Somali and Afar regions, while the lowest distribution was in the northern part of Ethiopia and in the Oromia region. For more detail, see (**Fig 2**).

**Hot spot analysis and interpolation of zero dose vaccination status.** As the Getis-Ord statistic showed, the significant hot spot area of zero-dose vaccination status among children in Ethiopia was located in Afar, Somalia, and SNNPR, whereas the northern part of Ethiopia was the significant cold spot area. For more detail, see **Fig 3**. To predict zero-dose vaccination status in an unsampled area, we used the spatial ordinary kriging method. Thus, it showed that

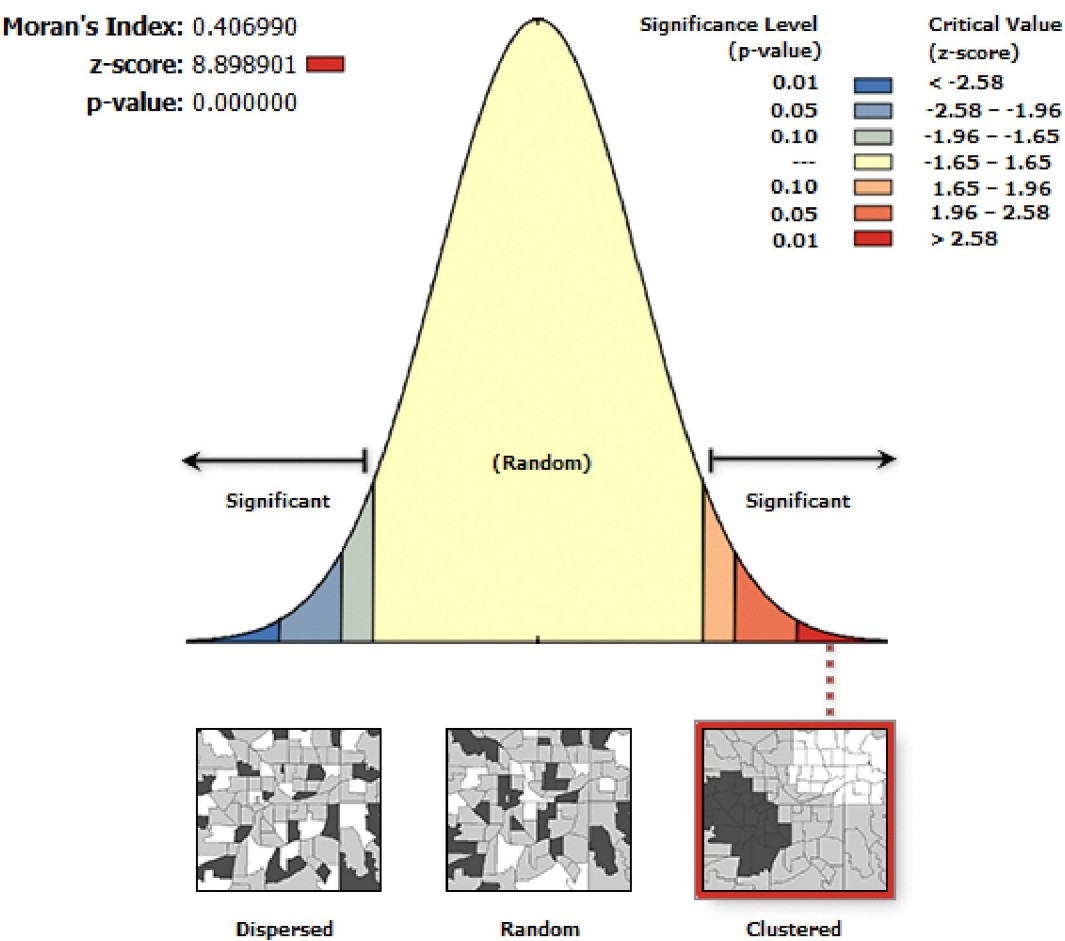

**Fig 1. Spatial auto-correlation of zero dose vaccination status among children in Ethiopia using EDHS 2019.**

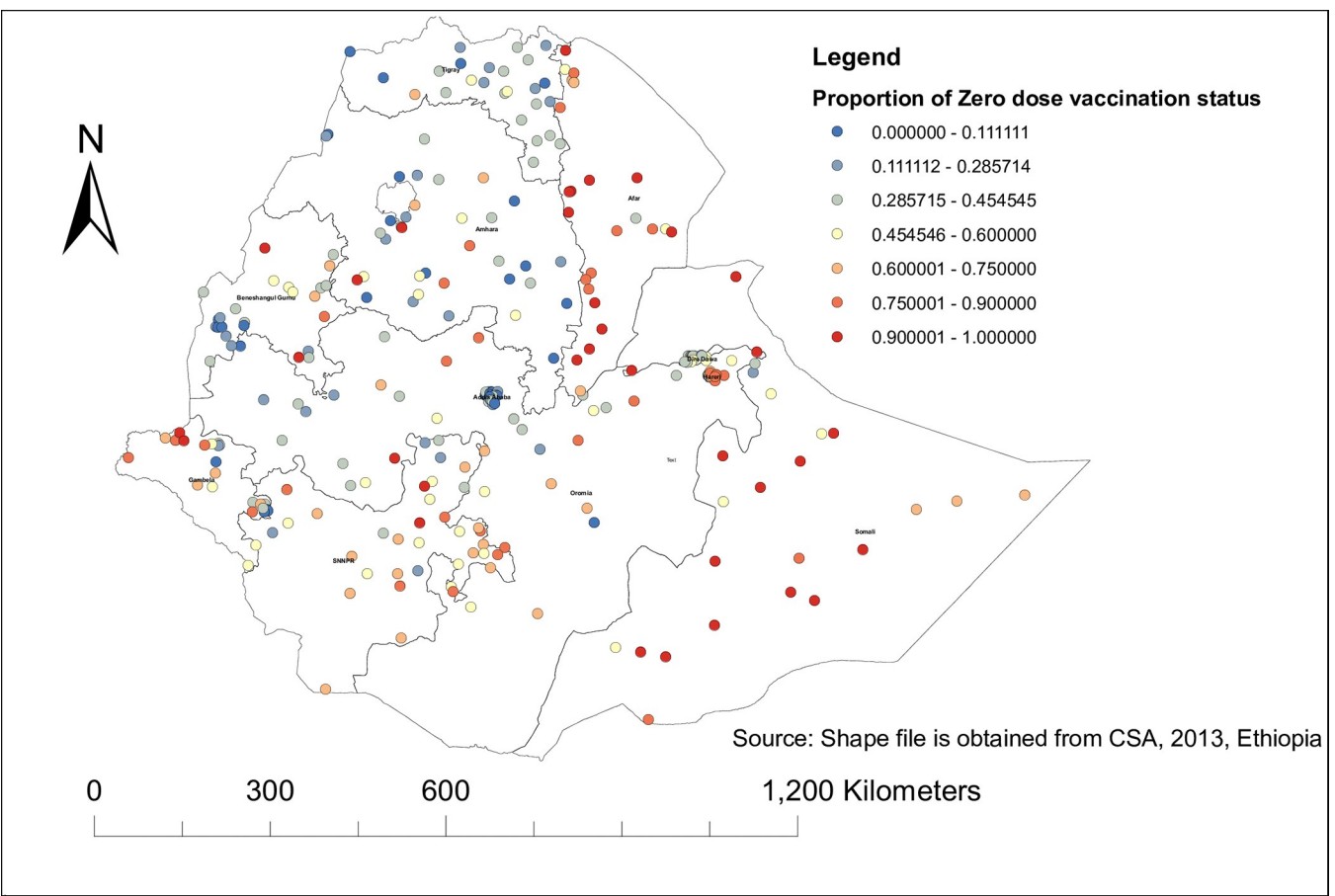

**Fig 2. Spatial distribution of zero dose vaccination status among children in Ethiopia using EDHS 2019.**

the highly predicted utilization of zero-dose vaccination status in Ethiopia was in Afar, Somalia, and the eastern part of Oromia, whereas the lower predicted area of zero-dose vaccination status in Ethiopia was in Addis Ababa and the western part of Benishangul Gumuz. For more detail, see **Fig 4**.

**Incremental autocorrelation of zero dose vaccination status.** The number of bands for the incremental autocorrelation was 10, and as shown, the band started at 150 kilometers. Which implies that at a distance of 150 kilometers (km) with a significant z-score value, the spatial clustering of zero-dose vaccination status was highly pronounced (**Fig 5**).

**SaT Scan analysis of zero dose vaccination status.** From the total of 305 clusters, 93 clusters were statistically significant. As stated in **Fig 6**, the blue-coloured window represented the primary clusters of the zero-dose vaccination status. Among the 93 clusters, 26 were the most likely (primary clusters). The primary cluster was located at 5.479641 N and 42.196835 E within a 405.96 km radius in the Somali region. In the most likely clusters, a zero-dose vaccination status was 1.63 times higher than outside the window (LLR = 69.89, RR = 1.63, p<0.001). For more detail, see **Table 2** and **Fig 6**.

**Factors affecting the spatial variation of zero dose vaccination status.** We used predictors like educational status, wealth index, ANC visit, place of delivery, and residence for analysis, and then home delivery and poor wealth index were the significant factors for zero-dose vaccine status (**Table 3**). Before we mapped and identified the factors affecting the spatial

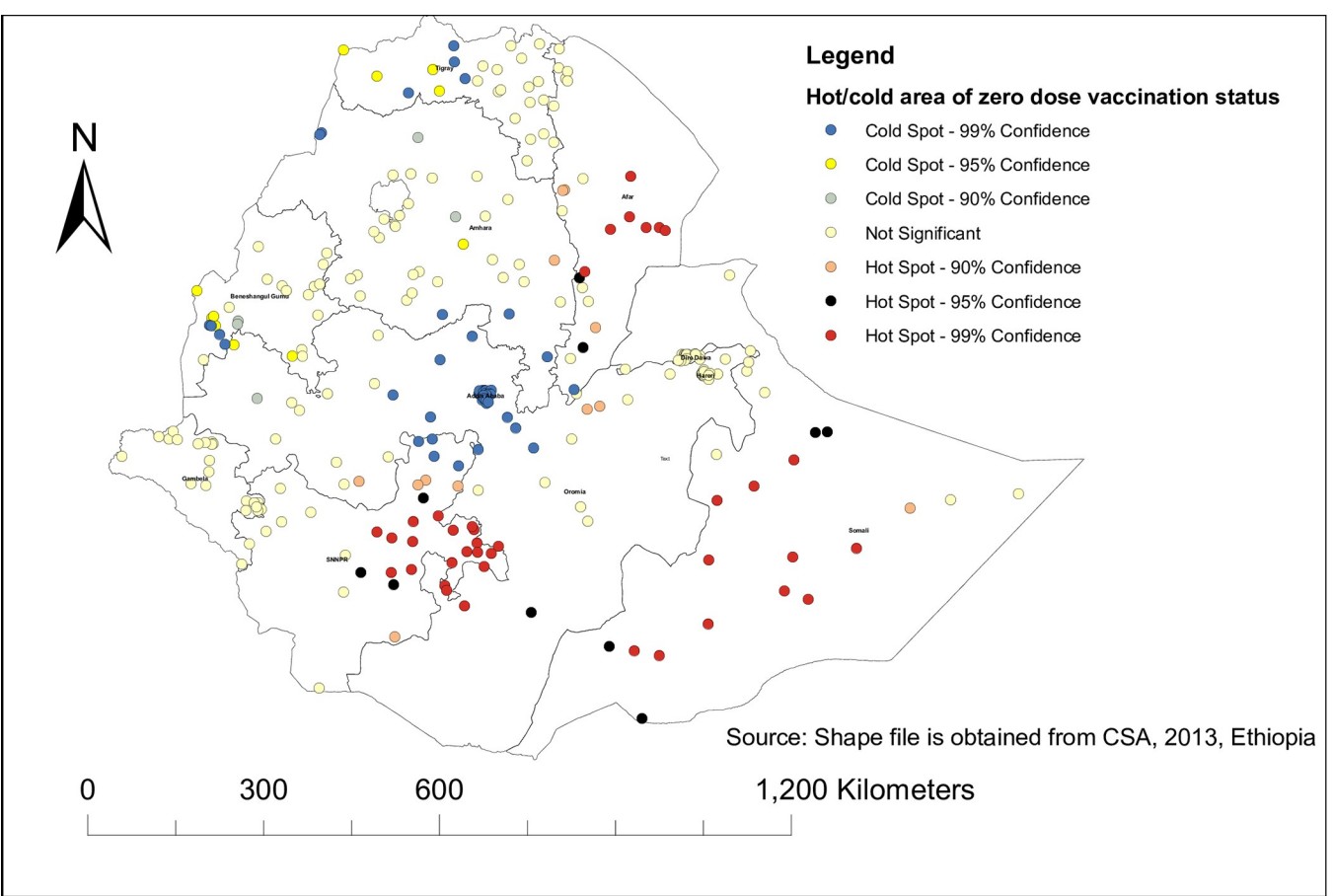

**Fig 3. Hot/cold spot area of zero dose vaccination status among children in Ethiopia using EDHS 2019.**

variation of zero-dose vaccination status, the necessary assumptions [39] were checked, and these assumptions were fulfilled. The normality of residuals was assessed by Jarque-Bera statistics (p-value = 0.476), and the multicollinearity was assessed by less than 7.5. The residual autocorrelation was evaluated by Moran's I statistics and revealed that it was not spatially correlated with a p-value of >0.05, and the model performance was evaluated by Akaike information criteria and $R^2$ change. Thus, the adjusted $R^2$ for OLS regression was 46%, whereas it was improved to 57.8% in GWR regression, and the AIC value was reduced from -508.50 in OLS to -679.20 in GWR (**Table 4**).

**Geographic weighted regression of zero dose vaccination status.** Based on the assumption of non-stationary and best model performance criteria, we used GWR (local method of analysis) to map the predictors of zero-dose vaccination status among children in Ethiopia. In general, the local method of analysis (GWR) improves the current model as compared to OLS regression. Thus, poor wealth index and home delivery were the positive predictors of zero-dose vaccination status. As the proportion of women with home deliveries increased, the zero-dose vaccination status increased. The highest effect (β, 0.23–0.24) of home delivery on zero dose vaccination status was observed in Afar, Somalia, and Tigray, whereas the lowest effect of home delivery (β, 0.21–0.22) was observed in SNNPR, Gambela, and the western part of Benishangul Gumuz (**Fig 7**). Furthermore, the proportion of women with poor wealth index increased the risk of zero dose vaccination status (not vaccinate their child), and as represented

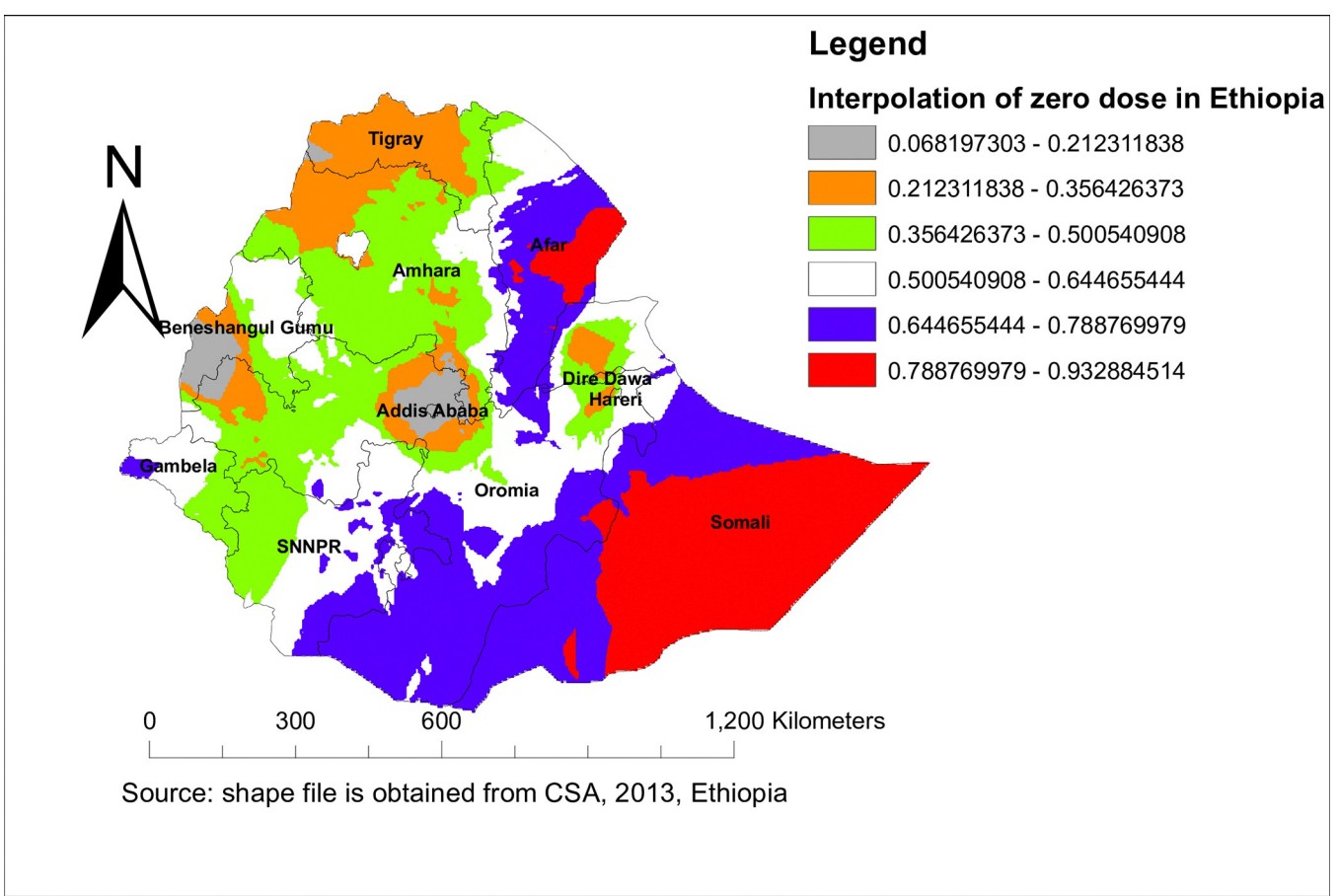

**Fig 4. Spatial interpolation of zero dose vaccination status among children in Ethiopia using EDHS 2019.**

by the red dot, the poor wealth index had the highest effect on zero dose vaccination status (β, 0.06–0.07) in SNNPR, Benishangul Gumuz, and the Gambela region, and the lowest effect of the poor wealth index (β, 0.03–0.04) on zero dose vaccination status was observed in Somalia and Afar (**Fig 8**).

## Discussion

The uptake of all basic vaccines is a pillar for the reduction of child mortality and morbidity from VPDs. But especially developing countries, including Ethiopia, are faced with the non-uptake of basic vaccines. Because of this, VPDs are a challenge for many poor countries. Even though it is still a public health problem, the magnitude, the spatial distribution, and the determinants of zero-dose vaccine status among children are not well known. Particularly, to apply interventions and resources to the hot spot of the problem, geospatial mapping of the problem and its determinants is very important. Therefore, in the current study, an attempt has been made to assess the spatial distribution of zero-dose vaccination status among children and apply geographical weighted regression to map the effect of each determinant across the area or region of Ethiopia.

The prevalence of zero-dose vaccination status among children aged 12–35 months old was 16.3%, (95% CI: 15%–17.6%). Even though the age group of the study population was varied, this finding was in line with a recent study conducted in Sub-Saharan Africa (16.5%) [1] and

**Spatial Autocorrelation by Distance**

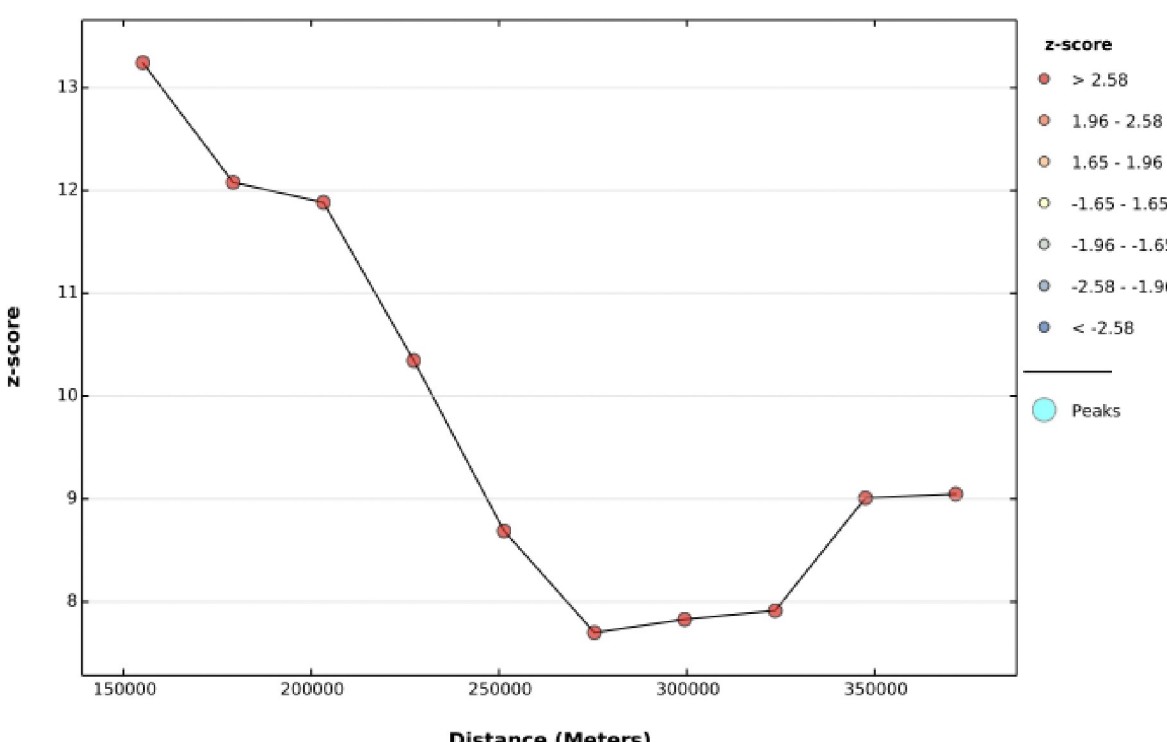

**Fig 5. Incremental autocorrelation of zero dose vaccination status among children in Ethiopia using EDHS 2019.**

the EDHS 2016 report (16%) [40]. But it was lower than another study conducted in Ethiopia by 33.7% (95% CI: 34.9%, 75.7%) [41]. This segment of the population is suffering from a lack of basic health services, including immunization. This causes a higher prevalence of zero-dose vaccination status among children. Additionally, the zero dose child in this study is lower than a study conducted between 2010 and 2020 using demographic health survey data set in sub-Saharan Africa (9.1%) [23]. The time variation, socioeconomic, and cultural differences of the study setting may make a difference. But the current study was higher than the study done in the 92 low- and middle-income countries (7.7%), with an average of 11.1% of children in low-income countries having not received any vaccines [11]. The possible reason might be associated with the variation in socioeconomic, health access, and health service uptake behaviour between the study settings.

The spatial distribution of zero-dose vaccination status in Ethiopia was non-randomly distributed across the regions and clusters, with a global Moran's index statistics value of 0.4 (p-value <0.001). Regarding the regional variations in zero-dose vaccination status, the highest proportion was reported in the Afar and Somali regions. Again, the hotspot area of zero-dose vaccination status was also in the Somali Afar and north-east parts of SNNPR. This might be because these regions are pastoralist. In the pastoralist region of Ethiopia, the health service utilization behavior of the women is very low [42]. Additionally, the majority of people's lives in the Somali Region and Afar Region are characterized by seasonal mobility, which negatively affects the utilization of health services [43]. People in the pastoralist regions also have low access to services and infrastructure, live in a traditional setting, and are influenced by cultural

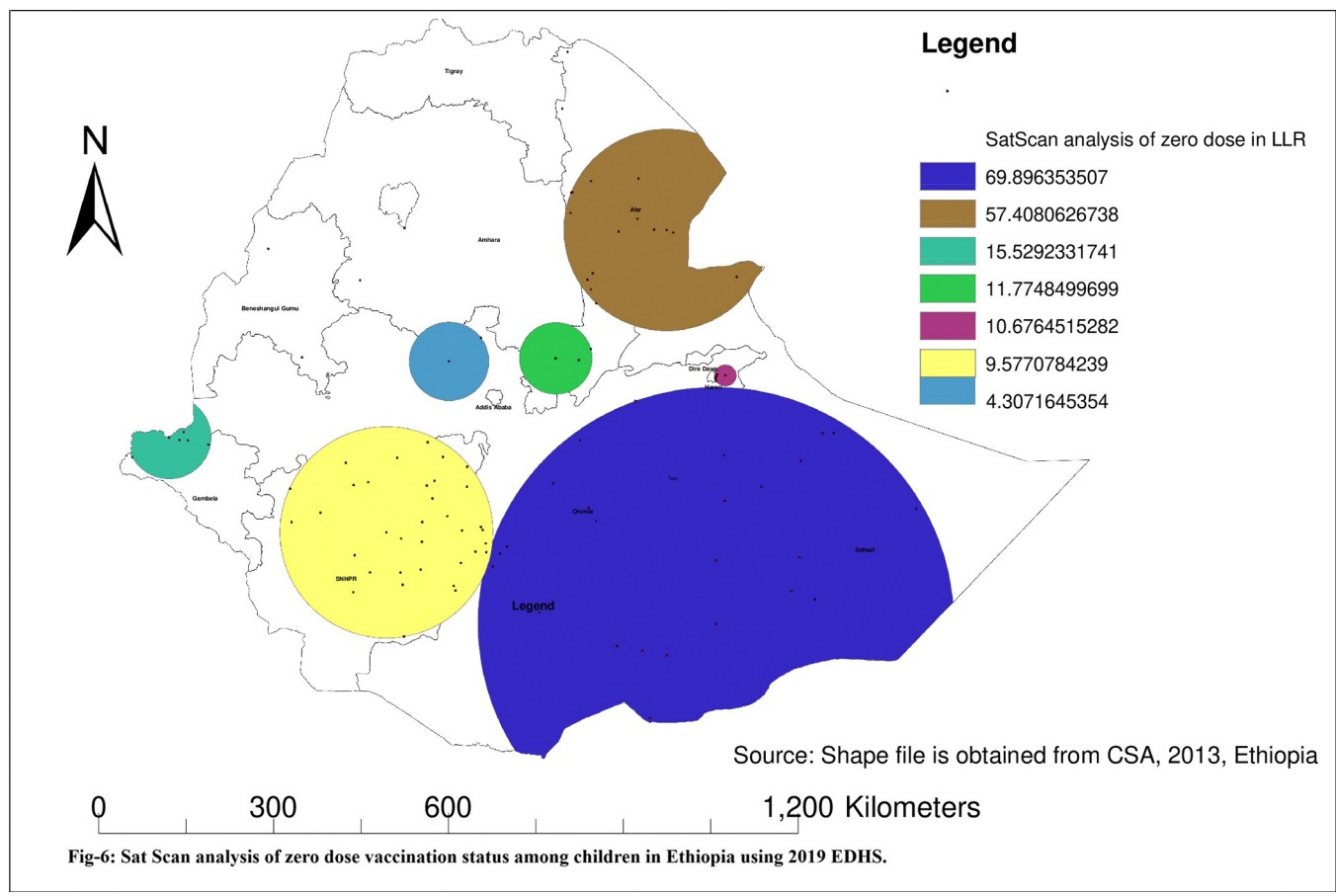

Fig-6: Sat Scan analysis of zero dose vaccination status among children in Ethiopia using 2019 EDHS.

**Fig 6. Sat Scan analysis of zero dose vaccination status among children in Ethiopia using 2019 EDHS.**

and religious values [44]. The significant primary cluster was located at 5.479641 N and 42.196835 E within a 405.96 km radius in the same region of Somali. In this area, a zero-dose vaccination status was 1.63 times higher than outside the area (LLR = 69.89, RR = 1.63,

**Table 2. Significant clusters of zero dose vaccination status among children in Ethiopia using EDHS 2019.**

| Cluster type | Significant Enumeration Areas (clusters) detected | Coordinate/radius | Population | Cases | RR | LLR | P-value |
|---|---|---|---|---|---|---|---|
| Primary | 142, 141, 136, 125, 138, 143, 137, 123, 144, 134, 145, 111, 135, 133,110, 114, 131, 103, 122, 117, 132, 183, 102, 113, 140, 106 | (5.479641 N, 42.196835 E) / 405.96 km | 348 | 288 | 1.63 | 69.89 | <0.001 |
| Secondary | 26, 32, 31, 30, 33, 34, 126, 47, 45, 48, 49, 44, 29, 46, 50 | (11.558430 N, 41.440210 E) / 172.92 km | 199 | 176 | 1.77 | 57.40 | <0.001 |
| Tertiary | 220, 217, 218, 229, 230, 219 | (8.356129 N, 33.766046 E) / 70.96 km | 79 | 66 | 1.56 | 15.50 | <0.001 |
| Quarterly | 69, 42, 40 | (9.577575 N, 39.728117 E) / 61.50 km | 35 | 32 | 1.70 | 11.77 | 0.0023 |
| Fifthly | 107, 254, 255, 249, 248, 250 | (9.312848 N, 42.343386 E) / 18.39 km | 89 | 69 | 1.45 | 10.68 | 0.0063 |
| Sixthly | 204, 191, 190, 189, 196, 198, 173, 197, 91, 199, 180, 96, 179, 178, 195, 192, 177, 184, 95, 115, 97, 172, 188, 182, 203, 176, 187, 201, 185, 174, 181, 186, 205, 194, 202 | (6.892369 N, 37.116362 E) / 181.26 km | 360 | 234 | 1.23 | 9.58 | 0.02 |
| Seventhly | 99, 100 | (9.531226 N, 38.081685 E) / 67.38 km | 20 | 17 | 1.57 | 4.31 | 0.03 |

**Table 3. Summary of OLS results for of zero dose vaccination status among children in Ethiopia using EDHS 2019.**

| Variable | Coefficient | Standard error | t-Statistic | Probability | Robust_SE | Robust_t | Robust_Pr | VIF |
|---|---|---|---|---|---|---|---|---|
| Intercept | 0.04 | 0.08 | 0.53 | 0.59 | 0.09 | 0.47 | 0.64 | —————— |
| Rural | 0.01 | 0.02 | 0.61 | 0.54 | 0.02 | 0.68 | 0.49 | 2.63 |
| Home delivery | 0.23 | 0.02 | 8.05 | <0.001* | 0.02 | 8.63 | **<0.001** | 2.75 |
| No education | -0.01 | 0.03 | -0.33 | 0.74 | 0.03 | 0.43 | 0.66 | 3.58 |
| Primary education | 0.01 | 0.04 | 0.31 | 0.76 | 0.03 | 0.407901 | 0.68 | 2.04 |
| Poor wealth index | 0.06 | 0.03 | 1.99 | 0.05* | 0.03 | 1.97 | **0.04*** | 3.42 |
| Middle wealth index | -0.06 | 0.04 | -1.34 | 0.18 | 0.04 | -1.36 | 0.18 | 1.72 |
| No ANC | -0.03 | 0.08 | -0.37 | 0.72 | 0.09 | -0.31 | 0.76 | 1.02 |

p<0.001). This is also the same reason for the living style and access to health services in the pastoralist area [43, 44]. pastoralist regions have low access to services and infrastructure, live in a traditional setting, and are influenced by cultural and religious values [44].

It was also found that poor wealth index and home delivery were the factors contributing to zero-dose vaccination status among children in Ethiopia. As the proportion of women who deliver at home increases, the zero dose vaccination status among children increases. This finding was supported by a study conducted in Cameroon [45] and a study conducted in sub-Saharan Africa. [1]. This might be because women who deliver at home are disadvantaged in the counselling of vaccination schedules and other related information. Home delivery had the highest effect (β, 0.23–0.24) on zero-dose vaccination status in Afar, Somalia, and Tigray, whereas the lowest effect of home delivery (β, 0.21–0.22) was observed in SNNPR, Gambela, and the western part of Benishangul Gumuz. This implies that the uptake of health services, including delivery services, across the regions of Ethiopia is variable, and its effect significantly influences the uptake of basic vaccines. Furthermore, as the proportion of women with a poor wealth index increased, the risk of zero-dose vaccination status (not vaccinating their child) increased. This was supported by a study conducted in low- and middle-income countries [46]. This might be associated with women with low socioeconomic status having a lower acceptability of health-related messages and a lower understanding of the vaccination advantage [47]. The effect of the poor wealth index was higher (β, 0.06–0.07) in SNNPR, Benishangul Gumuz, and the Gambela region, and the lowest effect of the poor wealth index (β, 0.03–0.04) on zero-dose vaccination status was observed in Somalia and Afar. This implies that the low wealth index varies across the regions of Ethiopia, and its effect significantly influences the

**Table 4. OLS Diagnosis and model comparison of zero dose vaccination status among children in Ethiopia using EDHS 2019.**

| **OLS Diagnosis** | | | |
|---|---|---|---|
| Numbers-of observation | 305 | Akaike information criteria(AIC) | -508.50 |
| Multiple $R^2$ | 47.20% | Adjusted $R^2$ | 46% |
| Joint F-statistics | 38 | Probability (>F), (7,297) degree | <0.001 |
| Joint Wald statistics | 264.90 | Probability(>chi-square), (7) degree of freedom | <0.001 |
| Koenker (BP) statistics | 47.90 | Probability(>chi-square), (7) degree of freedom | <0.001 |
| Jarque-Bera statistics | 10.69 | Probability(>chi-square), (2) degree of freedom | 0.479 |
| **Model comparison** | | | |
| Parameters | | OLS model | MGWR model |
| | AIC | -508.50 | -679.20 |
| | Multiple $R^2$ | 47.20% | 58.40% |
| | Adjusted $R^2$ | 46% | 57.80% |

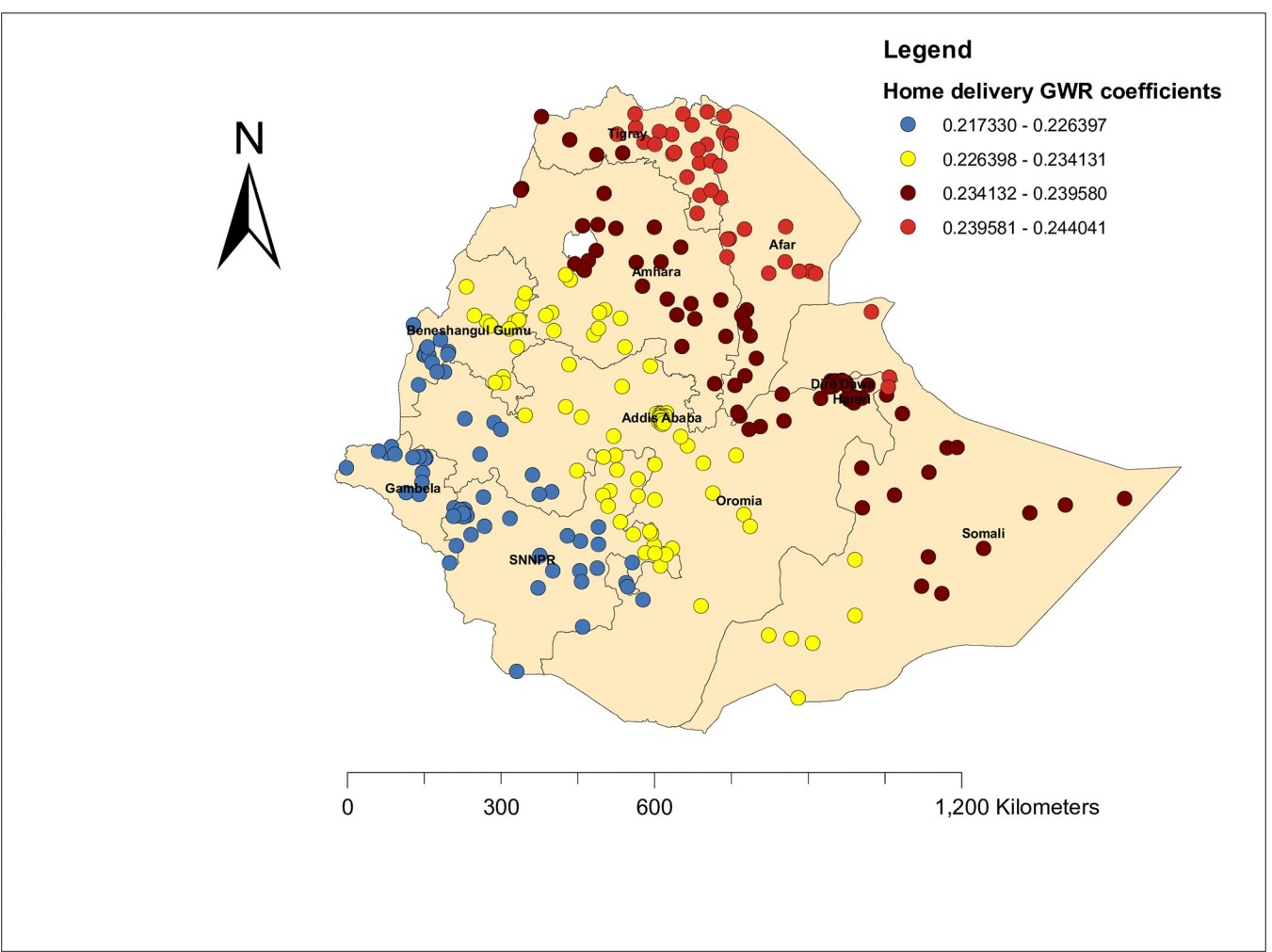

**Fig 7. Home delivery GWR coefficients of zero dose vaccination status among children in Ethiopia.**

uptake of basic vaccines. Alternatively, socioeconomic status may be counted as a more vital indicator for vaccine uptake as it has a more direct relationship with the uptake of vaccination with wealth index status [48].

These findings make it very important to prioritize the expansion of delivery services in health facilities and empower women to promote the uptake of childhood vaccines in Ethiopia. The strength of the study was using a nationally representative sample to identify the hot spots of the problem and the effects of each predictor across the areas to overcome to apply a limited resource in the prioritized areas. But it has limitations, such as the fact that the current study using the geographic coordinates of clusters (2 kilometers for urban areas and 5 kilometers for most clusters in rural areas) makes it difficult to estimate the cluster effect in the spatial analysis. Furthermore, the lack of studies with the same age range, make it more difficult to compare the results to earlier findings.

## Conclusion

The zero dose child in Ethiopia was low. Geographically, the highest proportion of zero-dose vaccination status among children was in the Somali and Afar regions. Home delivery had the

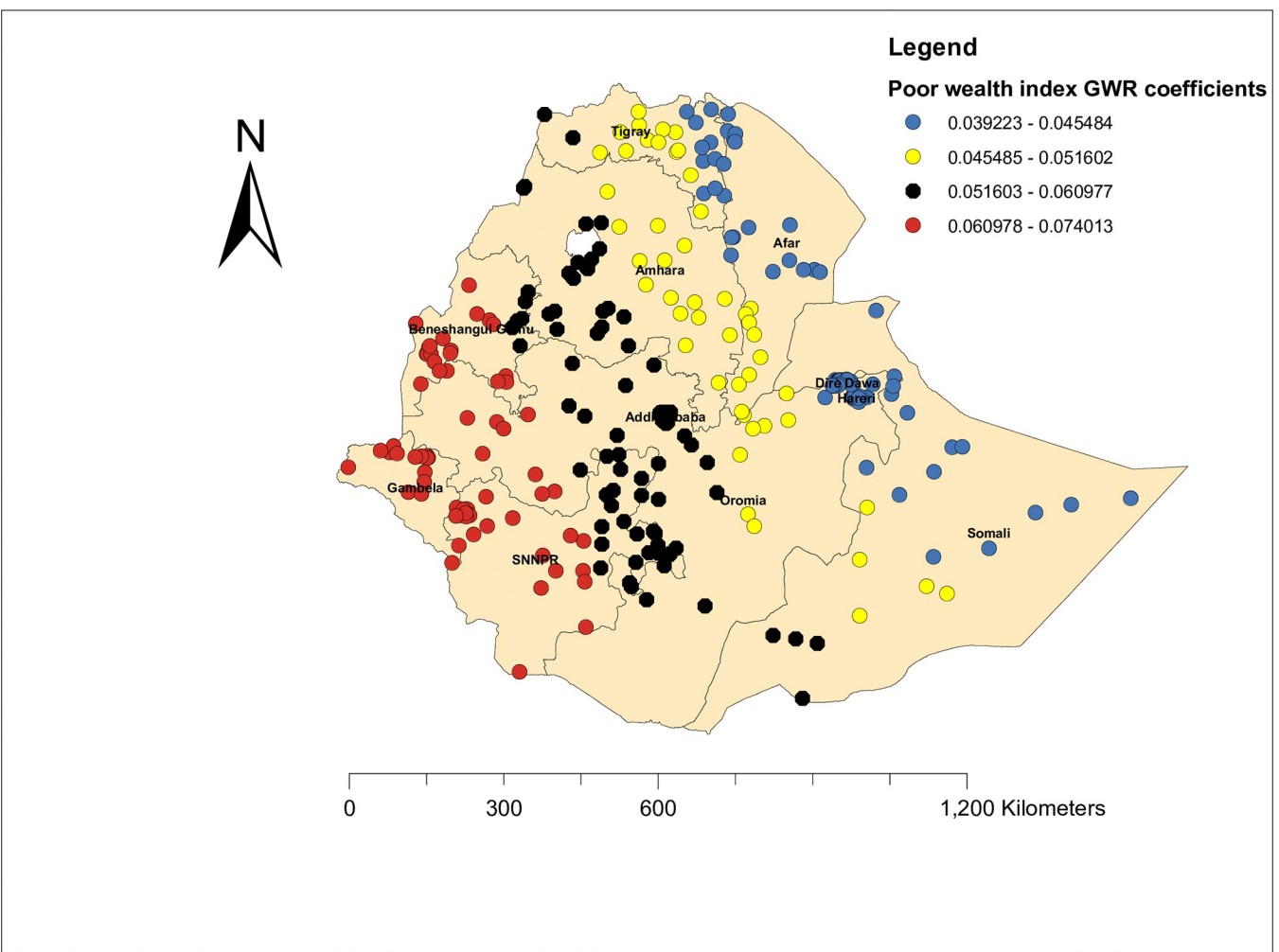

**Fig 8. Poor wealth index GWR coefficients of zero dose vaccination status among children in Ethiopia.**

highest and most positive effect on zero-dose vaccination status in Afar, Somalia, and the Tigray region. Additionally, the poor wealth index had the highest and most positive effect in Somalia and the Afar region. Expanding delivery at health facilities and economic empowerment of women are recommended to improve vaccination in Ethiopia.

## Supporting information

**S1 Checklist. STROBE statement—checklist of items that should be included in reports of observational studies.**
(DOCX)

**S1 Data.**
(RAR)

## Acknowledgments

The authors would like to give thanks to DHS International for accessing the data.

## Author Contributions

**Conceptualization:** Muluken Chanie Agimas, Meron Asmamaw, Nebiyu Mekonnen Derseh.

**Data curation:** Muluken Chanie Agimas, Meron Asmamaw, Tigabu Kidie, Habtamu Wagnew Abuhay, Nebiyu Mekonnen Derseh.

**Formal analysis:** Muluken Chanie Agimas, Meron Asmamaw, Tigabu Kidie, Habtamu Wagnew Abuhay, Getaneh Awoke Yismaw, Nebiyu Mekonnen Derseh.

**Investigation:** Muluken Chanie Agimas, Habtamu Wagnew Abuhay, Nebiyu Mekonnen Derseh.

**Methodology:** Muluken Chanie Agimas, Meron Asmamaw, Tigabu Kidie, Habtamu Wagnew Abuhay, Getaneh Awoke Yismaw, Nebiyu Mekonnen Derseh.

**Software:** Muluken Chanie Agimas, Tigabu Kidie, Nebiyu Mekonnen Derseh.

**Supervision:** Muluken Chanie Agimas, Nebiyu Mekonnen Derseh.

**Validation:** Muluken Chanie Agimas, Meron Asmamaw.

**Visualization:** Muluken Chanie Agimas, Meron Asmamaw, Melese Kebede Hailu, Tigabu Kidie.

**Writing – original draft:** Muluken Chanie Agimas, Meron Asmamaw, Melese Kebede Hailu, Tigabu Kidie, Nebiyu Mekonnen Derseh.

**Writing – review & editing:** Muluken Chanie Agimas, Meron Asmamaw, Melese Kebede Hailu, Tigabu Kidie, Nebiyu Mekonnen Derseh.

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
