## [Decision Letter · Decision Letter 0]

5 Jul 2024

PONE-D-24-01734Geospatial mapping to assess the distribution and determinants of zero dose vaccination status hot spots among children in Ethiopia using EDHS 2019: Spatial and geographical weighted regressionPLOS ONE

Dear Dr. Agimas,

Thank you for submitting your manuscript to PLOS ONE. After careful consideration, we feel that it has merit but does not fully meet PLOS ONE’s publication criteria as it currently stands. Therefore, we invite you to submit a revised version of the manuscript that addresses the points raised during the review process.

We look forward to receiving your revised manuscript.

Kind regards,

Balew Arega Negatie, Msc,MD

Academic Editor

PLOS ONE

3. In the online submission form you indicate that your data is not available for proprietary reasons and have provided a contact point for accessing this data. Please note that your current contact point is a co-author on this manuscript. According to our Data Policy, the contact point must not be an author on the manuscript and must be an institutional contact, ideally not an individual. Please revise your data statement to a non-author institutional point of contact, such as a data access or ethics committee, and send this to us via return email. Please also include contact information for the third party organization, and please include the full citation of where the data can be found.

4. We note that Figures 2, 3, 4, 6, 7 and 8 in your submission contain [map/satellite] images which may be copyrighted. All PLOS content is published under the Creative Commons Attribution License (CC BY 4.0), which means that the manuscript, images, and Supporting Information files will be freely available online, and any third party is permitted to access, download, copy, distribute, and use these materials in any way, even commercially, with proper attribution. For these reasons, we cannot publish previously copyrighted maps or satellite images created using proprietary data, such as Google software (Google Maps, Street View, and Earth). For more information, see our copyright guidelines: http://journals.plos.org/plosone/s/licenses-and-copyright.

1. You may seek permission from the original copyright holder of Figures 2, 3, 4, 6, 7 and 8 to publish the content specifically under the CC BY 4.0 license. 

Reviewers' comments:

Reviewer's Responses to Questions

**Comments to the Author**

1. Is the manuscript technically sound, and do the data support the conclusions?

Reviewer #1: Partly

Reviewer #2: Yes

2. Has the statistical analysis been performed appropriately and rigorously? 

Reviewer #1: Yes

Reviewer #2: Yes

3. Have the authors made all data underlying the findings in their manuscript fully available?

Reviewer #1: Yes

Reviewer #2: Yes

4. Is the manuscript presented in an intelligible fashion and written in standard English?

Reviewer #1: Yes

Reviewer #2: Yes

5. Review Comments to the Author

Reviewer #1: Overall

• The study focused on a significant and up-to-date subject concerning the health of children globally, with a specific emphasis on unvaccinated children (zero-dose) in Ethiopia. The authors utilized a valuable spatial approach to identify the hotspots for zero-dose status among children and its contributing factors. However, the introduction section lacks important information on national EPI vaccine coverages and estimates in the African countries.

• The authors should review the definition of zero-dose status among children.

• The authors should clarify their selection of children aged 12 to 35 months, given that numerous studies refer to children aged 12 to 23 months or 12 to 59 months, which would enhance comparability issues with other publications.

• The authors should consider expanding the discussion section with more studies conducted in the African region and recently published.

• The authors should review the conclusion

• The authors should review the manuscript for grammatical errors.

While the introduction section clearly articulates the burden of zero-dose around the world, there is insufficient information about the status of the national Expanded Program on Immunization including the current coverage rates, and the progress and challenges, if any, It will be important to understand whether there are notable complex contextual issues that may contribute to the current status of the national EPI and zero-dose prevalence.

Furthermore, numerous research conducted in the African region (DRC, Nigeria, etc.) have been published in recent months/years. These studies have examined the prevalence of zero-dose and identified the elements that contribute to it. The findings of these studies can be effectively utilised to support the argument for this study.

Lines 23-24; 67

Zero-dose vaccine status refers to a child who has not yet received any childhood vaccines. The authors could check on this definition.

According to Gavi, Zero-dose children are those that have not received any ROUTINE vaccine. For operational purposes, Gavi defines zero-dose children as those who lack the first dose of diphtheria-tetanus-pertussis containing vaccine (DTP1).

Line 25 “one child not access the vaccines” : one child did not access the vaccines

Lines 25-26 “The efforts of identifying the factors of the zero-dose vaccine yet not addressed in Ethiopia”: The efforts to identify the factors contributing to the zero-dose vaccine have not yet been addressed in Ethiopia.

Lines 34-35 To identify significant factors, a p-value <0.05 with 95% confidence was used.

Repetition. No need to present p-value and confidence level at the same time. They are complementary: “To identify significant factors, a p-value <0.05 was used” or “To identify significant factors, a 95% confidence was used”.

Lines 36-37 The prevalence of zero-dose vaccination status among children aged 12-35 months old was 523 (16.3%, 95%CI, 15%–17.6%). The authors could clarify on this sentence, I don’t understand it. What is 523? Prevalence is a proportion.

Line 36, 114 Question: Why age range of 12 to 35 months ?

Line 41 Question: Home delivery of what ? please precise

Lines 47-48 As compared to the previous studies, the prevalence of zero-dose vaccination status was low in Ethiopia This conclusion is not correct, 16.3% is a high prevalence rate compared to average prevalence varying from 5.2% in upper-middle income countries to 11.1% in low-income countries, for a total pooled prevalence of 7.7%.

Line 77-78 An increment of a zero-dose vaccine could be an indicator of not achieving the national vaccine coverage goal in sub-Saharan Africa (SSA) (90%) An increment of a zero-dose vaccine could be an indicator of not achieving the national vaccine coverage goal OVER TIME in sub-Saharan Africa (SSA) (90%)

Line 84 Question: Who are zero-dose population? Are they different from zero-dose children? Please be consistent in the use terms.

Lines 119 -120 The authors should distinguish independent variable related to household, to the woman and to the child

Lines 119 – 129; 125-127 While authors provided with the operational definition of “media exposure”, nothing was said regarding other independents variables.

Lines 119-120: Religion is quoted twice

Lines 208 - 209 zero dose vaccination status among children in Ethiopia was clustered (none randomly distributed)

Repetition. Clustered means not randomly distributed

The authors should consider expanding the discussion section with more studies conducted in the African region and recently published.

Lines 296 – 299 The authors compared the prevalence rate to those in Togo and Cameroon, but the three studies' definitions of zero-dose children and age ranges differed. The study in Togo defined zero-dose children as those who had not received the first dose of the diphtheria, tetanus, or pertussis vaccine, based on their immunization card, and their age ranged from 12 to 23 months. Similarly, the study in Cameroon defined zero-dose children as those who had not received any recommended vaccine antigen for their age, based on the Cameroon EPI calendar, and their age ranged from 0 to 24 months. The authors must take into account the comparability issues raised by the difference in their arguments.

Did the EDHS data rely on vaccination cards or maternal recall as the primary sources of information regarding childhood immunisation?

The authors should clarify the primary source used for this study

Lines 346 – 347 16% is a high prevalence not low. The pooled prevalence rate of zero-dose children is 7.7% in LMIC, with an average of 11.1% in Low-income countries (Cata-Preta BO, 2021) in a systematic review including 92 countries. The studies referred to here with high prevalence rates have a very specific context that has been explained, and they cannot be considered flagship studies for the prevalence of zero-dose children.

Reviewer #2: The article is very interesting, it presents a scientific and methodological rigor, with adequate statistical analysis to respond to the main objective of the study.

I suggest some corrections so that the text achieves the excellence required in this journal:

The abstract has many acronyms.

The introduction is based on an updated review of the scientific literature.

The methodology is well explained. I only suggest briefly explaining what the EDHS2019 study is and how it accounted for the ANC visit variable.

The results are well presented but I suggest some corrections:

Table 2 shows the 95% confidence interval of the RR, open tables 3 and 4 on the sides, and have homogeneity of decimal places in all tables.

The Discussion is backing up the results.

6. PLOS authors have the option to publish the peer review history of their article (what does this mean?). If published, this will include your full peer review and any attached files.

Reviewer #1: **Yes: **Armand Malembe Mutwadi

Reviewer #2: **Yes: **Lourdes Conceição Martins

---

## [Decision Letter · Decision Letter 1]

29 Sep 2024

PONE-D-24-01734R1Geospatial mapping to assess the distribution and determinants of zero dose vaccination status hot spots among children in Ethiopia using EDHS 2019: Spatial and geographical weighted regressionPLOS ONE

Dear Dr.Muluken

Thank you for submitting your manuscript to PLOS ONE. After careful consideration, we feel that it has merit but does not fully meet PLOS ONE’s publication criteria as it currently stands. Therefore, we invite you to submit a revised version of the manuscript that addresses the points raised during the review process.

We look forward to receiving your revised manuscript.

Kind regards,

Balew Arega Negatie, Msc,MD

Academic Editor

PLOS ONE

Journal Requirements:

Reviewers' comments:

Reviewer's Responses to Questions

**Comments to the Author**

1. If the authors have adequately addressed your comments raised in a previous round of review and you feel that this manuscript is now acceptable for publication, you may indicate that here to bypass the “Comments to the Author” section, enter your conflict of interest statement in the “Confidential to Editor” section, and submit your "Accept" recommendation.

Reviewer #1: (No Response)

Reviewer #2: All comments have been addressed

2. Is the manuscript technically sound, and do the data support the conclusions?

Reviewer #1: Partly

Reviewer #2: Yes

3. Has the statistical analysis been performed appropriately and rigorously? 

Reviewer #1: Yes

Reviewer #2: Yes

4. Have the authors made all data underlying the findings in their manuscript fully available?

Reviewer #1: No

Reviewer #2: Yes

5. Is the manuscript presented in an intelligible fashion and written in standard English?

Reviewer #1: Yes

Reviewer #2: Yes

6. Review Comments to the Author

Reviewer #1: Data availability statement: The authors did not provide a link to access the data or specify how to do so.

Line 151-152 “A two-stage cluster sampling method was used to select 3208 women (participants) who had children aged 12-35 months in the EAs” and Line 209 “Of the total 3028 women who had children aged 12-35 months old” : the author should fix the real sample size.

Line 217 : “The prevalence of zero-dose vaccination status among children aged 12-35 months old was 523” : prevalence is a proportion; the authors should add the denominator.

Line 310 – 312 : The current study does not have the same age range as the referral study (1), so the author should be cautious when interpreting the findings. What effect might a narrow age range have on the current research's prevalence when compared to a study with a wider age range?

Line 315-317 What is lower here ? your prevalence or the one conducted between 2010 and 2020. Please clarify.

Line 366 “the lack of comparable studies makes it more difficult to compare with the previous findings” : ‘The lack of studies with the same age range, makes it more difficult to compare the results to earlier findings’ this sounds more specific.

Line 46 and 368 “Ethiopia did not achieve the target of vaccination coverage” : Is this study intended to evaluate vaccine coverage? Which target of vaccine coverage are you talking about here? Did you calculate the vaccine coverage in this study? For which antigen? The research findings, in my opinion, do not support this statement.

Reviewer #2: (No Response)

7. PLOS authors have the option to publish the peer review history of their article (what does this mean?). If published, this will include your full peer review and any attached files.

Reviewer #1: **Yes: **Armand M. Mutwadi

Reviewer #2: No

---

## [Editor Report · Decision Letter 2]

10 Oct 2024

Geospatial mapping to assess the distribution and determinants of zero dose vaccination status hot spots among children in Ethiopia using EDHS 2019: Spatial and geographical weighted regression

PONE-D-24-01734R2

Dear Dr.Muluken Chanie

We’re pleased to inform you that your manuscript has been judged scientifically suitable for publication and will be formally accepted for publication once it meets all outstanding technical requirements.

Kind regards,

Balew Arega Negatie, Msc,MD

Academic Editor

PLOS ONE
---

## [Editor Report · Acceptance letter]

14 Oct 2024

PONE-D-24-01734R2 

PLOS ONE

Dear Dr. Agimas, 

I'm pleased to inform you that your manuscript has been deemed suitable for publication in PLOS ONE. Congratulations! Your manuscript is now being handed over to our production team.

Kind regards, 

on behalf of

Dr. Balew Arega Negatie 

Academic Editor

PLOS ONE